# Dietary Arginine Supplementation Improves Intestinal Mitochondrial Functions in Low-Birth-Weight Piglets but Not in Normal-Birth-Weight Piglets

**DOI:** 10.3390/antiox10121995

**Published:** 2021-12-15

**Authors:** Hao Zhang, Ping Zheng, Daiwen Chen, Bing Yu, Jun He, Xiangbing Mao, Jie Yu, Yuheng Luo, Junqiu Luo, Zhiqing Huang, Hui Yan

**Affiliations:** 1Animal Nutrition Insititute, Sichuan Agricultural University, Chengdu 611130, China; zh199412062021@163.com (H.Z.); chendwz@sicau.edu.cn (D.C.); Ybingtian@163.com (B.Y.); hejun8067@163.com (J.H.); acatmxb2003@163.com (X.M.); yujie@sicau.edu.cn (J.Y.); luoluo212@126.com (Y.L.); junqluo2018@tom.com (J.L.); zqhuang@sicau.edu.cn (Z.H.); yan.hui@sicau.edu.cn (H.Y.); 2Key Laboratory for Animal Disease-Resistance Nutrition, China Ministry of Education, Chengdu 611130, China

**Keywords:** low birth weight, piglets, arginine, jejunum, mitochondrial functions, redox status

## Abstract

Our previous studies revealed that _L_-arginine supplementation had beneficial effects on intestinal barrier functions of low-birth-weight (LBW) piglets, which were associated with the enhanced antioxidant capacity. Moreover, mitochondrial functions are closely related to the redox state. This study was to explore potential mechanisms of _L_-arginine-induced beneficial effects against intestinal dysfunction by regulating mitochondrial function of LBW piglets. Twenty 4-day-old normal birth weight (NBW) piglets (BW: 2.08 ± 0.09 kg) and 20 LBW siblings (BW: 1.16 ± 0.07 kg) were artificially fed either a basal diet or a basal diet supplemented with 1.0% _L_-arginine for 21 d, respectively. Growth performance, intestinal morphology, redox status, mitochondrial morphology, and mitochondrial functions were examined. Data were subjected to two-way analysis of variance. LBW piglets presented lower (*p* < 0.05) ADG, shorter (*p* < 0.05) intestinal villus height, lower (*p* < 0.05) jejunal adenosine triphosphate (ATP) content and higher (*p* < 0.05) concentrations of Ca^2+^ and 8-OH-dG in jejunal mitochondria, compared with NBW piglets. Supplementation with 1.0% _L_-arginine significantly increased (*p* < 0.05) ADG, the activities of CAT, SOD, and GPx, intestinal villus height and mRNA abundances of *ZO-1* (2-fold) in the jejunum of LBW piglets, but not in NBW piglets. Furthermore, the concentrations of ATP and the transcription of *COX IV*, *COX V* genes were up-regulated (*p* < 0.05) and the concentration of Ca^2+^ and 8-OH-dG were decreased (*p* < 0.05) in arginine-treated LBW piglets. The results suggest that mitochondrial morphology is affected, and mitochondrial functions are impaired in the jejunum of LBW piglets. While supplementation with 1.0% _L_-arginine relieved intestinal dysfunction through enhancing antioxidant capacity and improving mitochondrial functions via repairing mitochondrial morphology, normalizing mitochondrial calcium, and increasing ATP concentration in the jejunum of LBW piglets. However, supplementation with _L_-arginine has no significant beneficial effects on intestinal health in NBW piglets.

## 1. Introduction

Low birth weight (LBW) is a tough problem with human health and animal production [1,2]. LBW neonates show higher rates of perinatal mortality and have long-term adverse effects on postnatal growth, intestinal development and health [3,4,5]. The small intestine is the dominating organ for digestion and absorption of nutrients [6]. Gastrointestinal dysplasia and dysfunction are two of the most important reasons for the retarded growth of LBW neonates [7]. A previous study has revealed that impaired antioxidant capacity results in intestinal dysfunction and the retarded growth of LBW suckling piglets, which is closely associated with impaired intestinal health [5,7,8]. Meanwhile, LBW is associated with increased prevalence of metabolic disorders that are closed to mitochondria dysfunctions [9]. Mitochondria are considered to be a source of reactive oxygen species (ROS) [10]. Recently, the focus has been more directed towards mitochondrial dysfunction as a potential mechanism that impairs redox balance [11]. Studies have shown that mitochondrial dysfunction is proven by the lower ATP synthesis and decreased levels of mitochondria function-related genes, such as electron transport chain process, oxidative phosphorylation and mitochondrial biogenesis in LBW neonates, which is associated with the decreased activities of antioxidant enzymes and excess production of ROS [4,12,13,14]. Moreover, previous studies showed that mitochondria played a critical role in cell apoptosis, and LBW neonates are likely to occur cell apoptosis [15,16]. Moreover, an increase in antioxidant capacity might relieve oxidative stress and enhance the functions of mitochondria [17]. Therefore, attenuating mitochondrial dysfunction may be an efficient strategy to maintain intestinal development and functions in LBW neonates.

Arginine is an essential amino acid for young mammals [18,19]. Several studies revealed that supplementation with arginine had access to improve growth performance and enhance the antioxidant capacity of pigs [5,20,21,22]. Studies also reported that supplementing arginine enhanced intestinal development in 7–21-day-old NBW piglets [23,24,25,26]. Our previous research found that supplementation with arginine improved intestinal barrier functions via enhanced antioxidant capacity of 4–25-day-old LBW piglets [5]. However, little is known about the difference in the benefits of arginine supplementation in intestinal development between NBW and LBW offspring. Moreover, arginine has the functions to promote the biogenesis of mitochondria and attenuate mitochondrial dysfunction [27,28]. Therefore, further investigations are required for clarifying the underlying mechanism of arginine supplementation on intestinal mitochondrial function of LBW offspring.

Pigs (*Sus scrofa*) share a high similarity in gastrointestinal physiology with humans. LBW piglets had been considered as an ideal model to study gastrointestinal syndrome in LBW infants [29]. Thus, studying the effects of arginine supplementation on mitochondrial functions and intestinal health of LBW piglets could reflect that of corresponding human phenotypes. Based on the aforementioned findings, we hypothesized that arginine supplementation could relieve mitochondrial dysfunction to enhance antioxidant capacity for better intestinal function and greater growth performance in LBW piglets. In the present study, we evaluated the differences in mitochondrial function between NBW and LBW piglets supplemented with _L_-arginine in terms of growth performance, intestinal morphology, redox status, mitochondrial function-related gene expression. This study is significant for further understanding the mechanism of the arginine-induced beneficial effects on intestinal health in LBW offspring.

## 2. Materials and Methods

The experiment was conducted in accordance with the guidelines of the Ethics Committee of Sichuan Agricultural University (Yaan, China) for the use of animals in research.

### 2.1. Experimental Animals and Diets

Referring to our previous study [5], LBW piglets were defined as newborns with birth weights less than 1.0 kg and NBW piglets were defined as newborns with birth weight within 1 SD unit of the mean birth weight of the whole litter. Newborn piglets were weighed within 2 h after parturition (day 114 ± 1 of gestation) to determine the birth weight (BW). A total of twenty LBW and twenty NBW crossbred piglets (Large White × Landrace; Chengdu, China) were chosen from 20 sows in this study. Each litter provided 1 LBW piglet and 1 NBW piglet. On day 4, half of the NBW and LBW piglets were fed control milk replacer (CON), and the other half were fed 1% _L_-arginine supplementation milk replacer (ARG) through bottle-feeding. The average BW of LBW and NBW piglets on day 4 was 1.16 (±0.07) kg and 2.08 (±0.09) kg, respectively. Thus, a 2 × 2 factorial design was adopted with 10 piglets per treatment (NBW-CON, NBW-Arg, LBW-CON, and LBW-Arg), and the sex ratio was equal (1:1).

The basal diet was formulated according to our previous study [5], as shown in Appendix A. The liquid diets were prepared by mixing 1.0 kg formula powder (dry matter: 87.5%) with 4 L warm water (40 °C).

The experimental management was the same as our previous study [5]. Briefly, all piglets were housed separately in metabolism cages (0.8 m × 0.7 m × 0.4 m) in an environmentally controlled room. The initial room temperature was maintained at 31 ± 1 °C for the first week and was gradually decreased to 28 ± 1 °C by the end of the experiment. Piglets were given *ad libitum* access to water and bottle-fed with liquid formula milk. The piglets were bottle-fed 7 times per day at 06.00, 09.00, 12.00, 15.00, 18.00, 21.00, and 24.00 h for the whole experimental period. The daily milk dry matter intake for each piglet was calculated according to the weight of liquid milk intake. No medicines or antibiotics were used during the experimental period. The experiment lasted 21 days.

### 2.2. Sample Collection

All piglets were weighed after an overnight fast on day 22 of the experiment. Then, all piglets were sampled (*n* = 10). The piglets were euthanized by intravenous injection of pentobarbital sodium (50 mg/kg of BW), then killed approved by the Sichuan Agricultural University Animal Care Advisory Committee, and the abdomens were immediately incised for the collection of gut samples. The jejunum samples were collected as described in our previous study [30]. Briefly, a 10 cm segment of the jejunum was emptied, carefully flushed with saline, and placed on an ice-cold surface. The mucosa was gently scraped with a glass slide and transferred to a plastic 2 mL screw-capped tube, snap-frozen in liquid nitrogen, and then stored at −80 °C for further analyses. Then, a 2 cm segment of jejunum was collected, trimmed into a 1 mm^3^ square and fixed in 2.5% glutaraldehyde solution to detect the morphology of mitochondria. Subsequently, 2 cm segment of mid-jejunum was collected, flushed with ice-cold PBS (pH 7.4) and then fixed in 10% fresh, chilled formaldehyde solution for histomorphologic measurements.

### 2.3. Assay of Serum Insulin

Serum insulin concentration was determined by immunoassays using a reagent kit (Jiangsu Jingmei Biotechnology Co., Ltd., Yancheng, China) according to the manufacturer’s instructions.

### 2.4. Assay of Enzyme Activity and ATP Concentration

Jejunal mucosa samples were homogenized as previously described [5]. Then, the supernatant was transferred into Eppendorf tubes and stored at −80 °C for further analyses. Activities of superoxide dismutase (SOD), glutathione peroxidase (GPX), and catalase (CAT) and concentrations of malondialdehyde (MDA), adenosine triphosphate (ATP), and calcium ion (Ca^2+^) in jejunal mucosa were determined through enzymatic colorimetric methods according to commercial SOD, GPX, CAT, MDA, ATP, and Ca^2+^ assay kits (Nanjing Jiancheng Bioengineering Institute, Nanjing, China).

### 2.5. Analysis of Jejunal Morphology

The morphology of the jejunum was measured as described previously [31]. Briefly, after fixing in 10% formaldehyde solution, the jejunum was embedded in paraffin. The consecutive section (5 μm) was stained with hematoxylin–eosin. Then, villus height and crypt depth were determined with an Olympus CK 40 microscope (Olympus Optical Company, Shenzhen, China). Ten intact villi and associated crypts were randomly selected in each sample and measured.

### 2.6. Transmission Electron Microscopy

The morphology of jejunal mitochondria was measured as described previously [32]. Briefly, the sample was fixed in 2.5% glutaraldehyde, rinsed in 0.1 M sodium phosphate buffer (pH 7.2) twice and then fixed in osmic acid for 3 h. After dehydration, the sample was embedded and sectioned into ultrathin sections. Then sections were observed with a transmission electron microscope (Philips, Eindhoven, The Netherlands).

### 2.7. Assay of Mitochondrial MDA, 8-OH-dG, and mPTP

Mitochondria were extracted from jejunal mucosa. Briefly, jejunal mucosa samples were homogenized in a 0 °C separation medium (1:9 wt:vol, including sucrose 0.25 mol/L, mannitol 0.05 mol/L, EDTA 1 mmol/L, and Tris 10 mmol/L, pH 7.5), centrifuged at 600× *g* for 7 min at 4 °C. Then, the supernatant was transferred into Eppendorf tubes, centrifuged at 1600× *g* for 5 min at 4 °C. Next, the supernatant was transferred into Eppendorf tubes, centrifuged at 12,500× *g* for 10 min at 4 °C, and the remaining sediment is the mitochondria. The sediment was suspended with a 5 mL separation medium and then centrifuged at 12,500× *g* for 5 min at 4 °C to obtain the sediment of purified mitochondria.

Malondialdehyde (MDA) in jejunal mitochondria was determined through enzymatic colorimetric methods according to commercial MDA assay kits (Nanjing Jiancheng Bioengineering Institute). 8-hydroxydeoxyguanosine (8-OH-dG) and permeability transition pore (mPTP) concentrations in jejunal mitochondria were determined by commercially available porcine-specific ELISA kits (Jiangsu Jingmei Biotechnology Co., Ltd.) and an automatic biochemical instrument (Biochemical Analytical Instrument, Beckman CX4, Beckman Coulter Inc., Brea, CA, USA). All measurements were conducted in triplicate at a minimum according to the manufacturer’s instructions.

### 2.8. RNA Isolation and Real Time-qPCR

Total RNA was isolated from the jejunal mucosa samples by using RNAiso Plus reagent (TaKaRa, Dalian, China) according to the manufacturer’s instructions. The concentration of RNA in the final preparations was calculated from the OD260/280. The integrity of RNA was verified using denaturing agarose gel electrophoresis. Reverse transcription with the use of the Prime Script RT reagent kit (TaKaRa, Dalian, China) was operated according to the manufacturer’s instructions.

Specific primers for Zona Occludens 1 (*ZO1*), Occludin, Claudin 1, Peroxisome proliferator-activated receptor-gamma coactivator-1α (*PGC-1α*), Mitochondria transcriptional factor A (*TFAM*), Nuclear respiratory factor 1(*NRF1*), Cytochrome c oxidase I (*CoxI*), Cytochrome c oxidase IV (*CoxIV*), Cytochrome c oxidase V (*CoxV*), Cytochrome C (*Cyt C*), fission 1(*FIS1*), Dynamin-related protein 1 (*Drp1*), Optic atrophy 1 (*OPA1*), mitofusin 1(*MFN 1*), and mitofusin 2 (*MFN2*)were designed and purchased from Invitrogen (Shanghai, China), which are listed in Appendix A.

Real-time quantitative PCR was performed in an Option Monitor 3 Real-Time PCR Detection System (Bio-Rad, Hercules, CA, USA) using the SYBR Green Supermix (TaKaRa, Dalian, China). The PCR system consisted of 10 μL SYBR Premix Ex Taq II, 2 μL cDNA template, 1 μL forward primer, 1 μL reverse primer, and 6 μL ultrapure water. Cycling conditions were as follows: a precycling at 95 °C for 30 s, 40 cycles at 95 °C for 5 s, and annealing at optimal temperature for 30 s, and 1 cycle at 95 °C for 15 s. The melting curve was conducted following each real-time quantitative PCR to check and verify the specificity of all PCR products. The β-actin gene was chosen as the reference gene to normalize the mRNA expression of target genes [33]. The relative gene expression was calculated by the 2^−ΔΔCT^ method [34].

### 2.9. Statistical Analyses

All analyses were performed using two-way ANOVA in SPSS version 22.0 software package (IBM, SPSS Statistics for Windows, New York, NY, USA). Normality of data and homogeneity of variance were tested with the use of Levene’s test. All results were applied to the following model to analyze data:Y*ijk* = *μ* + *αi* + *βj* + (*αβ*)*ij* + *εijk* (*i* = 1, 2; *j* = 1, 2; *k* = 1, 2, …, *nij*)
where Y*ijk* represents the dependent variable; *μ* is the mean; *αi* is the effect of BW; *βj* is the effect of Arg; (*αβ*)*ij* is the interaction between the BW and Arg; and *εijk* is the error term.

When the main effect and/or interaction were significant by two-way ANOVA using a GLM procedure followed by Duncan’s test was performed. The sample size of each treatment was *n* = 10. The data are expressed as mean ± SD. A *p*-value < 0.05 was used to indicate a significant difference, and a *p*-value between 0.05 and 0.10 indicated a trend.

## 3. Results

### 3.1. Food Intake, BWs, and General Health

No diarrhea occurred in all of the piglets throughout the trial. The average daily dry matter intake (ADMI) of LBW piglets was less than that of NBW littermates, whereas ADMI in LBW piglets supplemented with 1.0% _L_-arginine was greater (*p* < 0.05) than that of LBW piglets in CON treatment (Table 1). Arginine intakes in arginine-treated NBW and LBW piglets were greater (*p* < 0.05) than those of control piglets.

Compared with NBW piglets, a significantly lower BW and ADG were observed in all LBW piglets (Table 2). However, a significantly higher RADG was found in LBW piglets. Meanwhile, dietary supplementation with 1.0% _L_-arginine increased (*p* < 0.05) ADG of 17% in LBW piglets compared with LBW CON piglets. Consequently, the BWs of the 25-d-old LBW piglets supplemented with 1.0% _L_-arginine were 13% greater (*p* < 0.05) than those of LBW CON piglets. However, supplementation with 1.0% _L_-arginine had no effect (*p* > 0.05) on ADG and final BW in NBW piglets.

### 3.2. Serum Insulin Concentration

As presented in Figure 1, supplementation with 1.0% _L_-arginine increased (*p* < 0.05) serum insulin concentration of LBW piglets but not in NBW piglets (*p* > 0.05).

### 3.3. Intestinal Morphology

As shown in Table 3, the villus height of LBW piglets that were fed the CON diet was lower (*p* < 0.05) than that of NBW piglets. Supplementation with 1.0% _L_-arginine increased (*p* < 0.05) the villus height and villus height/crypt depth in jejunum of LBW piglets.

### 3.4. Redox Status in the Jejunum

As shown in Table 4, supplementation with 1.0% _L_-arginine enhanced (*p* < 0.05) the activities of CAT, SOD, and GPx in LBW piglets but not in NBW piglets (*p* > 0.05).

### 3.5. Gene Expressions of Tight Junction Proteins

The relative mRNA expressions of *ZO-1* in the jejunum of LBW piglets were down-regulated (*p* < 0.05) when compared with NBW piglets (Table 5). The supplementation of 1.0% _L_-arginine increased (*p* < 0.05) the mRNA level of *ZO-1* in LBW piglets and *Claudin*
*1* in NBW piglets.

### 3.6. Mitochondrial Morphology

Mitochondrial morphology was given in Figure 2. Well-developed mitochondria with preserved membranes were found in NBW piglets (A). Supplementation with 1.0% _L_-arginine had no significant effects on the mitochondrial morphology in the jejunum of NBW piglets (B). However, the swelling and vacuole mitochondria in the jejunum were found in LBW piglets (C). Supplementation with 1.0% _L_-arginine could alleviate the degree of swelling and decrease the incidence of irregular-shaped mitochondria in the jejunum of LBW piglets (D).

### 3.7. Jejunal Free Ca^2+^, mPTP and ATP Concentration

Compared with NBW piglets, ATP concentration was significantly decreased in the jejunum of LBW piglets (Table 6). Supplementation with 1.0% _L_-arginine resulted in a lower (*p* < 0.05) Ca^2+^ concentration and higher (*p* < 0.05) ATP concentration in the jejunum of LBW piglets.

### 3.8. Mitochondrial 8-OH-dG and MDA Concentration

There was a significant increase in 8-OH-dG concentration in the jejunal mitochondria of LBW piglets when compared with NBW piglets (Table 7). Supplementation with 1.0% _L_-arginine decreased (*p* < 0.05) 8-OH-dG concentration in the jejunal mitochondria of LBW piglets.

### 3.9. Gene Expressions of Mitochondrial Biogenesis and Function in the Jejunum

Table 8 shows the gene expressions of mitochondrial biogenesis and function in the jejunum. The mRNA levels of *Cyt C*, *OPA1*, *Mfn1*, and *Mfn2* in the jejunum of LBW piglets were less than (*p* < 0.05) that of NBW piglets. Supplementation with 1.0% _L_-arginine resulted in higher (*p* < 0.05) *TFAM*, *COX IV*, *COX V*, *OPA1*, and *Mfn2* mRNA levels and lower *Fis1* mRNA levels in the jejunum of LBW piglets. Meanwhile, supplementation with 1.0% _L_-arginine exhibited higher *COX V*, *ATPS*, *OPA1*, and *Mfn2* mRNA levels in the jejunum of NBW piglets (*p* < 0.05).

## 4. Discussion

Our previous study indicated that LBW resulted in poor growth performance and intestinal dysfunction with lower antioxidant capacity in piglets, and _L_-arginine supplementation could improve intestinal development and enhance growth by reducing oxidative stress in LBW piglets [5]. The present study further investigated whether dietary arginine supplementation affected intestinal health through regulating mitochondrial functions of piglets with different birth weights. Using the model of artificially reared neonatal piglets, we demonstrated that dietary arginine supplementation improved intestinal morphology and enhanced antioxidant capacity and promoted growth performance by increasing ADMI and ADG in LBW piglets, but it had no significant effects on NBW piglets. Furthermore, dietary arginine supplementation improved mitochondrial functions and decreased the concentration of 8-Oh-dG in jejunal mitochondria of LBW piglets, but it had no significant effects on NBW piglets. These results indicate that supplementation with _L_-arginine may be beneficial only to LBW piglets. The possible reasons for the beneficial effects of _L_-arginine supplementation on LBW piglets are as follows.

Firstly, the arginine requirement of LBW piglets is higher than that of NBW piglets. Our previous study [5] found that the requirements for LBW piglets (1–7 kg body weight) were lysine: arginine (100:71) was higher than the recommended requirements (lysine: arginine 100:45 for 5–7 kg piglets) by NRC (2012) [35]. In the present study, we found that the growth performance of LBW piglets was lower than that of NBW piglets, but supplementation with 1.0% _L_-arginine could significantly improve the growth performance of LBW piglets. However, NBW piglets exhibited the same growth performance, whether _L_-arginine was supplemented or not. In this study, the ratio of lysine to arginine in the diet without _L_-arginine was 100:39, and the ratio of lysine to arginine in the diet with _L_-arginine was 100:85. Therefore, we suggested that high arginine level (lysine: arginine 100:85) had a better growth promotion effect on LBW piglets than low arginine level (Lysine: Arginine 100:39), and the level was higher than that recommended (Lysine: Arginine 100:45) by NRC(2012) [35]. Moreover, we also found RADG of LBW piglets was higher than that of NBW littermates, but dietary supplementation with 1.0% _L_-arginine still significantly increased RADG of LBW piglets while having no effect on RADG in NBW piglets. Moreover, Zheng (2013) et al. [20] found that supplementation with 0.8% or 1.6% _L_-arginine, to a certain extent, decreased ADG and ADFI of NBW weaned piglets. These results indicated that excessive addition of arginine has no improvement or even inhibition effect on the growth performance of piglets.

Secondly, the antioxidant function of arginine only takes effects when the body is subjected to oxidative stress. The first line defense antioxidants, including SOD, CAT, and GPX, are indispensable in the entire defense strategy of antioxidants [36]. Consistent with previous studies [6,20,37]), supplementation with 1.0% _L_-arginine was able to enhance SOD, CAT and GPx activities in the jejunum of LBW piglets. However, the results showed that supplementation with _L_-arginine had no effects on the antioxidant capability of NBW piglets. The possible reason why _L_-arginine increased the antioxidant capacity of LBW piglets but had no significant effect on NBW piglets was that LBW piglets were subjected to oxidative stress, while NBW piglets were not. Many studies have reported that LBW neonates suffer oxidative stress [7,38,39]. Our previous study showed that dietary supplementation of arginine effectively relieves oxidative stress by increasing the activities of GPx and SOD in plasma and liver of diquat-induced oxidative stress piglets [20]. However, dietary arginine had no effects on the activities of antioxidant enzymes of piglets with injection with sterile saline [20]. On the other hand, studies have shown that oxidative state is relative to intestinal morphology [5,40]. This study confirmed that intestinal morphology of LBW piglets was impaired, and tight junction protein gene expression was down-regulated. Conversely, supplementation with _L_-arginine improved the intestinal morphology of LBW piglets and up-regulated the expression of the *ZO-1* gene. Therefore, we speculate that dietary _L_-arginine supplementation can alleviate jejunal dysfunction by enhancing the antioxidant capacity of LBW piglets.

Thirdly, mitochondrial dysfunction is the possible underlying mechanism of redox imbalance [10]. Kowaltowski et al. [41] suggested that decreased mitochondrial energy production associated with oxidative stress could disturb mitochondrial function, leading to substantial swelling of the organelle. Adenosine triphosphate is the main energy source for maintaining cellular physiological responses [42]. In this study, ATP concentration was significantly lower in the jejunum of LBW piglets compared with NBW piglets. Previous research suggested that cellular energy supply was mainly controlled by mitochondrial biogenesis [43]. In line with this, we also found the relative genes expressions of mitochondrial biogenesis and oxidative phosphorylation were low in the jejunum of LBW piglets, which was consistent with the results of Liu et al.’s study [3] in the livers of LBW piglets. TFAM is a major regulator of mitochondrial DNA transcription and replication [44]. COX IV, COX V, and Cyt C are vital to mitochondrial respiratory chain membrane protein [45,46]. Moreover, OPA1, Mfn1 and Mfn2 are the key regulatory factors of mitochondrial fusion. Conversely, Drp1 and Fis1 are the regulation factors of mitochondrial fission [47]. These processes control not only the shape but also the mitochondrial function [48]. In the current study, the mRNA levels of *Cyt C*, *OPA1*, *Mfn1*, and *Mfn2* in the jejunum of LBW piglets were lower than those in NBW piglets, which demonstrated that mitochondrial dysfunction occurred in LBW piglets. Meanwhile, the jejunum Ca^2+^ concentration of LBW piglets was significantly higher than that of NBW piglets. Matrix Ca^2+^ overload is the primary trigger for the opening of the mPTP, resulting in mitochondrial swelling, decreased membrane potential, oxidative phosphorylation decoupling, and mitochondrial damage [49]. Therefore, lower ATP and Ca^2+^ overload may lead to mitochondria swelling and mitochondrial damage. As it turns out, jejunum mitochondria of LBW piglets were swollen, irregular, and fractured by electron microscopy observation in this study. Mitochondria are considered to be the source of ROS, and the reduced ability to eliminate ROS may trigger oxidative stress and impair mitochondrial function [10]. 8-Hydroxy-2′-deoxyguanosine (8-OH-dG) is one of the most abundant oxidative products of mtDNA [50] that is related to the proteins synthesis of the respiratory chain [51]. As shown in the results presented in this study, 8-OH-dG contents in the jejunum of LBW piglets were significantly higher than those of NBW piglets. Thus, mitochondrial dysfunction in LBW piglets may induce excessive accumulation of mitochondrial oxidation products, which subsequently triggers oxidative stress.

Finally, _L_-arginine can restore mitochondrial function in LBW piglets. In this study, we proved that supplementation with 1.0% _L_-arginine reduced mitochondrial Ca^2+^ overload and increased ATP concentration in the jejunum of LBW piglets. Meanwhile, supplementation with 1.0% _L_-arginine enhanced mitochondrial biogenesis and function-related gene expressions, such as *TFAM*, *COX IV*, *COX V*, *OPA1*, and *Mfn2* mRNA levels, in the jejunum of LBW piglets. Mitochondrial dysfunction has been linked to a number of diseases, including diabetes, asthma, and so on. Previous studies showed that exogenously administered _L_-arginine could improve mitochondrial function by restoring the normal electron transport chain in diabetic rats [52] and alleviate mitochondrial dysfunction in murine allergic airway inflammation [53]. In addition, the addition of 1.0% _L_-arginine decreased 8-OH-dG concentration, oxidative products of mtDNA, in the jejunum of LBW piglets. These effects were paralleled by improvement of mitochondrial morphology abrogating swollen and irregular-shaped mitochondria by arginine administration. Therefore, these results indicated that supplementation with 1.0% _L_-arginine may ameliorate mitochondrial dysfunctions through various pathways, including regulating mitochondrial biogenesis, oxidative phosphorylation, and mitochondrial fusion, in the jejunum of LBW piglets.

## 5. Conclusions

In conclusion, the results of this study provide evidence for impaired intestinal mitochondrial functions of the jejunum associated with reduced expressions of the tight-junction protein ZO-1 as well as increased concentrations of 8-OH-dG in LBW piglets. Meanwhile, supplementation with 1.0% _L_-arginine relieved intestinal dysfunction by enhancing the antioxidant capacity and attenuating mitochondrial dysfunction in LBW piglets. Furthermore, the beneficial effects of _L_-arginine on mitochondrial functions were mainly through various pathways, including improving mitochondrial biogenesis, oxidative phosphorylation, and mitochondrial fusion, in the jejunum of LBW piglets. Collectively, these changes may explain some mechanisms responsible for intestinal dysfunction in LBW neonates. Thus, the research provides an experiment base for deeply studying intestinal dysfunction in LBW piglets and offers some important clues for the development of novel nutrition strategies for LBW infants and piglets to attenuate postnatal retarded growth.

## Figures and Tables

**Figure 1 antioxidants-10-01995-f001:**
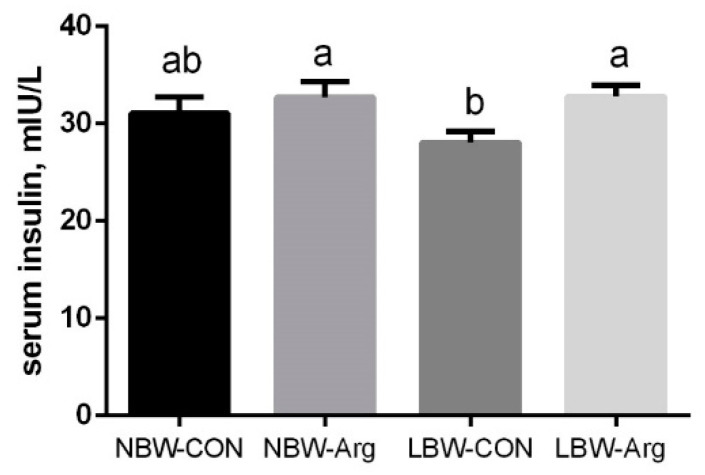
Effects of _L_-arginine supplementation on serum insulin concentration in suckling piglets with different birth weights. Values are expressed as mean with their SEM, *n* = 10/group. Means without a common letter differ, *p* < 0.05. Arg, a diet supplemented with 1.0% _L_-arginine; CON, a diet not supplemented with _L_-arginine; LBW, low birth weight; NBW, normal birth weight.

**Figure 2 antioxidants-10-01995-f002:**
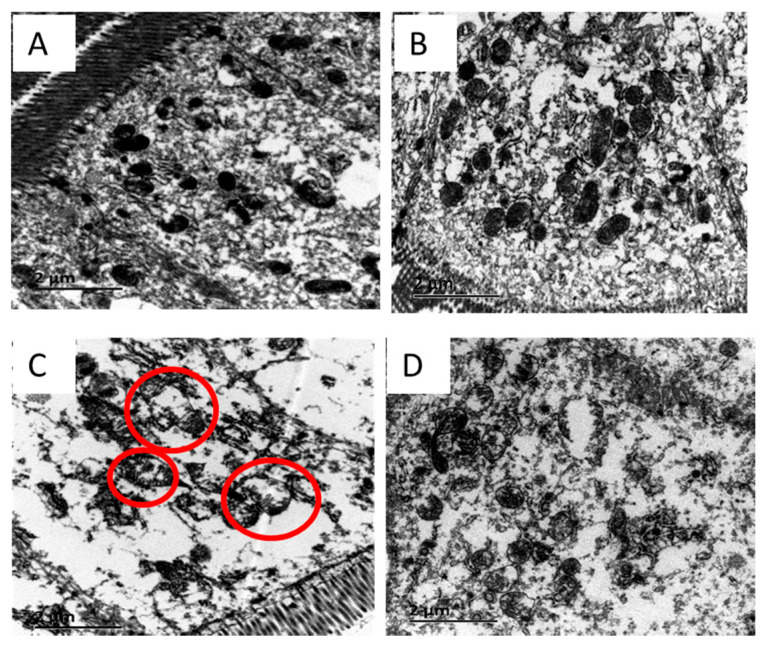
Effects of dietary _L_-arginine supplementation on jejunum mitochondrial morphology in piglets with different birth weights. (**A**), normal birth weight piglets fed a diet not supplemented with _L_-arginine; (**B**), normal birth weight piglets fed a diet supplemented with 1.0% _L_-arginine; (**C**), low birth weight piglets fed a diet not supplemented with _L_-arginine (red circle: the typical swollen and vacuolar mitochondria); (**D**), low birth weight piglets fed a diet supplemented with 1.0% _L_-arginine.

**Table 1 antioxidants-10-01995-t001:** Effects of _L_-arginine supplementation on relative and absolute food intakes in suckling piglets with different birth weights ^1^.

Parameter ^2^	NBW ^3^		LBW ^4^		*p*-Value		
	CON ^5^	Arg ^6^	CON	Arg	BW	Arg	BW × Arg
ADMI, g/d	229 ± 27 ^a^	244 ± 18 ^a^	144 ± 7 ^c^	174 ± 12 ^b^	0.000	0.003	0.273
RADMI, g·(kg BW^−1^·d^−1^)	42.75 ± 3.18 ^b^	44.10 ± 1.47 ^b^	43.37 ± 2.31 ^b^	46.91 ± 1.64 ^a^	0.051	0.007	0.202
AAI, g/d	1.95 ± 0.23 ^c^	4.51 ± 0.34 ^a^	1.23 ± 0.06 ^d^	3.21 ± 0.22 ^b^	0.000	0.000	0.004
RAI g·(kg BW^−1^·d^−1^)	0.36 ± 0.03 ^c^	0.82 ± 0.03 ^b^	0.37 ± 0.02 ^c^	0.87 ± 0.03 ^a^	0.006	0.000	0.022

^1^ Values are expressed as mean with their SEM, *n* = 10/group. Labeled means in a row without a common letter differ, *p* < 0.05. ^2^ ADMI, average daily dry matter intake; AAI, absolute arginine intake; RADMI, relative average daily dry matter intake; RAI, relative arginine intake; ^3^ NBW, normal birth weight; ^4^ LBW, low birth weight; ^5^ CON, a diet not supplemented with _L_-arginine; ^6^ Arg, a diet supplemented with 1.0% _L_-arginine.

**Table 2 antioxidants-10-01995-t002:** Effects of _L_-arginine supplementation on growth performance of suckling piglets with different birth weights ^1^.

Parameter ^2^	NBW ^3^		LBW ^4^		*p*-Value		
	CON ^5^	Arg ^6^	CON	Arg	BW	Arg	BW × Arg
Initial BW on day 4, kg	2.07 ± 0.10 ^a^	2.09 ± 0.09 ^a^	1.16 ± 0.09 ^b^	1.17 ± 0.08 ^b^	0.000	0.637	0.850
Final BW on day 25, kg	8.65 ± 0.85 ^a^	8.95 ± 0.53 ^a^	5.50 ± 0.24 ^c^	6.24 ± 0.56 ^b^	0.000	0.019	0.292
ADG, g/d	314 ± 37 ^a^	327 ± 25 ^a^	207 ± 9 ^c^	242 ± 15 ^b^	0.000	0.014	0.239
RADG, g·(kg BW^−1^·d^−1^)	151 ± 13 ^c^	157 ± 13 ^c^	179 ± 12 ^b^	208 ± 13 ^a^	0.000	0.001	0.020
FE, g gain/g feed	1.37 ± 0.11	1.34 ± 0.06	1.44 ± 0.08	1.39 ± 0.03	0.067	0.197	0.823

^1^ Values are expressed as mean with their SEM, *n* = 10/group. Labeled means in a row without a common letter differ, *p* < 0.05. ^2^ ADG, average daily gain; BW, body weight; FE, feed efficiency. RADG, relative average daily gain; ^3^ NBW, normal birth weight. ^4^ LBW, low birth weight. ^5^ CON, a diet not supplemented with _L_-arginine. ^6^ Arg, a diet supplemented with 1.0% _L_-arginine.

**Table 3 antioxidants-10-01995-t003:** Effect of _L_-arginine supplementation on jejunal morphology of suckling piglets with different birth weights ^1^.

Parameter	NBW ^2^		LBW ^3^		*p*-Value		
	CON ^4^	Arg ^5^	CON	Arg	BW	Arg	BW × Arg
Villus height, µm	848 ± 162 ^a^	845 ± 86 ^a^	608 ± 65 ^b^	935 ± 121 ^a^	0.116	0.002	0.002
Crypt depth, µm	159 ± 30	146 ± 34	172 ± 23	152 ± 19	0.384	0.139	0.725
Villus height/Crypt depth	5.56 ± 1.83 ^a^	6.01 ± 1.24 ^a^	3.58 ± 0.71 ^b^	6.22 ± 1.07 ^a^	0.105	0.007	0.049

^1^ Values are expressed as mean ± mean with their SEM, *n* = 10/group. Labeled means in a row without a common letter differ, *p* < 0.05. ^2^ NBW, normal birth weight. ^3^ LBW, low birth weight. ^4^ CON, a diet not supplemented with _L_-arginine. ^5^ Arg, a diet supplemented with 1.0% _L_-arginine.

**Table 4 antioxidants-10-01995-t004:** Effects of _L_-arginine supplementation on antioxidant enzymes activities and MDA concentrations in jejunum of suckling piglets with different birth weights ^1^.

Parameter ^2^	NBW ^3^		LBW ^4^		*p*-Value		
	CON ^5^	Arg ^6^	CON	Arg	BW	Arg	BW × Arg
MDA, nmol/mg protein	0.38 ± 0.23	0.31 ± 0.16	0.23 ± 0.07	0.26 ± 0.11	0.095	0.719	0.406
CAT, U/mg protein	21.53 ± 5.08 ^ab^	18.28 ± 3.16 ^b^	18.29 ± 3.33 ^b^	25.97 ± 6.60 ^a^	0.180	0.183	0.002
SOD, U/mg protein	134.73 ± 21.82 ^b^	133.94 ± 23.17 ^b^	149.61 ± 30.58 ^b^	177.67 ± 31.35 ^a^	0.003	0.146	0.125
GPx, U/mg protein	85.02 ± 22.59 ^ab^	91.55 ± 49.43 ^ab^	71.40 ± 15.04 ^b^	114.96 ± 29.68 ^a^	0.673	0.037	0.118

^1^ Values are expressed as mean with their SEM, *n* = 10/group. Labeled means in a row without a common letter differ, *p* < 0.05. CAT, catalase; ^2^ GPx, glutathione peroxidase; MDA, malondialdehyde; SOD, superoxide dismutase. ^3^ NBW, normal birth weight. ^4^ LBW, low birth weight. ^5^ CON, a diet not supplemented with _L_-arginine. ^6^ Arg, a diet supplemented with 1.0% _L_-arginine.

**Table 5 antioxidants-10-01995-t005:** Effects of _L_-arginine supplementation on relative mRNA expression of ZO1, Occludin and Claudin1 in the jejunum of LBW piglets ^1^.

Parameter ^2^	NBW ^3^		LBW ^4^		*p*-Value		
	CON ^5^	Arg ^6^	CON	Arg	BW	Arg	BW × Arg
*ZO-1*	1.00 ± 0.16 ^a^	1.25 ± 0.43 ^a^	0.59 ± 0.13 ^b^	1.05 ± 0.32 ^a^	0.005	0.001	0.296
*Occludin*	1.00 ± 0.35 ^ab^	1.35 ± 0.51 ^a^	0.74 ± 0.20 ^b^	1.11 ± 0.35 ^ab^	0.059	0.008	0.943
*Claudin 1*	1.00 ± 0.11 ^b^	1.50 ± 0.44 ^a^	0.93 ± 0.08 ^b^	1.02 ± 0.50 ^b^	0.020	0.013	0.082

^1^ Values are expressed as mean with their SEM, *n* = 10/group. Labeled means in a row without a common letter differ, *p* < 0.05. ^2^
*ZO-1*, Zona Occludens 1. ^3^ NBW, normal birth weight. ^4^ LBW, low birth weight. ^5^ CON, a diet not supplemented with L-arginine. ^6^ Arg, a diet supplemented with 1.0% L-arginine.

**Table 6 antioxidants-10-01995-t006:** Effects of _L_-arginine supplementation on free Ca^2+^, ATP, and mPTP concentrations in jejunum of suckling piglets with different birth weights ^1^.

Parameter ^2^	NBW ^3^		LBW ^4^		*p*-Value		
	CON ^5^	Arg ^6^	CON	Arg	BW	Arg	BW × Arg
Ca^2+^, mmol/g protein	0.19 ± 0.02 ^b^	0.19 ± 0.02 ^b^	0.21 ± 0.02 ^a^	0.19 ± 0.02 ^b^	0.372	0.112	0.049
mPTP, ng/g protein	52.09 ± 7.82	50.90 ± 12.87	54.06 ± 4.53	55.56 ± 3.88	0.256	0.976	0.592
ATP, μmol/g protein	154.37 ± 19.02 ^a^	152.69 ± 16.02 ^a^	106.44 ± 15.49 ^c^	135.68 ± 8.34 ^b^	0.000	0.017	0.008

^1^ Values are expressed as mean with their SEM, *n* = 10/group. Labeled means in a row without a common letter differ, *p* < 0.05. ^2^ ATP, adenosine triphosphate; Ca^2+^, calcium ion; mPTP, mitochondrial permeability transition pore. ^3^ NBW, normal birth weight. ^4^ LBW, low birth weight. ^5^ CON, a diet not supplemented with _L_-arginine. ^6^ Arg, a diet supplemented with 1.0% _L_-arginine.

**Table 7 antioxidants-10-01995-t007:** Effects of _L_-arginine supplementation on 8-OH-dG and MDA concentrations in jejunal mitochondria of suckling piglets with different birth weights ^1^.

Parameter ^2^	NBW ^3^		LBW ^4^		*p*-Value		
	CON ^5^	Arg ^6^	CON	Arg	BW	Arg	BW × Arg
8-OH-dG, ng/g protein	2.38 ± 0.59 ^b^	2.02 ± 0.26 ^b^	2.89 ± 0.48 ^a^	2.25 ± 0.33 ^b^	0.019	0.002	0.365
MDA, nmol/mg protein	4.30 ± 0.55	3.82 ± 0.50	4.04 ± 0.40	3.89 ± 0.81	0.651	0.120	0.396

^1^ Values are expressed as mean with their SEM, *n* = 10/group. Labeled means in a row without a common letter differ, *p* < 0.05. ^2^ MDA, malondialdehyde; 8-OH-dG, 8-hydroxydeoxyguanosine. ^3^ NBW, normal birth weight. ^4^ LBW, low birth weight. ^5^ CON, a diet not supplemented with _L_-arginine. ^6^ Arg, a diet supplemented with 1.0% _L_-arginine.

**Table 8 antioxidants-10-01995-t008:** Effects of _L_-arginine supplementation on mitochondrial biogenesis and function related genes expression in jejunum of suckling piglets with different birth weights ^1^.

Parameter ^2^	NBW ^3^		LBW ^4^		*p*-Value		
	CON ^5^	Arg ^6^	CON	Arg	BW	Arg	BW × Arg
*PGC-1α*	1.00 ± 0.32	1.17 ± 0.33	1.03 ± 0.15	1.13 ± 0.16	0.951	0.155	0.672
*TFAM*	1.00 ± 0.16 ^ab^	1.00 ± 0.13 ^ab^	0.86 ± 0.24 ^b^	1.15 ± 0.08 ^a^	0.903	0.014	0.015
*NRF1*	1.00 ± 0.29	1.03 ± 0.16	0.95 ± 0.26	1.06 ± 0.14	0.924	0.395	0.591
*COX I*	1.00 ± 0.27	1.19 ± 0.34	1.01 ± 0.21	0.90 ± 0.23	0.145	0.708	0.118
*COX IV*	1.00 ± 0.34 ^b^	1.04 ± 0.26 ^b^	0.95 ± 0.14 ^b^	1.37 ± 0.19 ^a^	0.183	0.160	0.309
*COX V*	1.00 ± 0.35 ^bc^	1.31 ± 0.30 ^a^	0.85 ± 0.13 ^c^	1.23 ± 0.13 ^ab^	0.216	0.001	0.701
*Cyt C*	1.00 ± 0.30	0.94 ± 0.26	0.73 ± 0.20	0.83 ± 0.29	0.040	0.891	0.277
*Fis1*	1.00 ± 0.26 ^b^	0.92 ± 0.47 ^b^	1.46 ± 0.52 ^a^	0.84 ± 0.16 ^b^	0.165	0.012	0.046
*Drp1*	1.00 ± 0.43 ^ab^	0.61 ± 0.42 ^b^	1.32 ± 0.57 ^a^	0.95 ± 0.61 ^ab^	0.064	0.033	0.964
*OPA1*	1.00 ± 0.15 ^b^	1.24 ± 0.30 ^a^	0.68 ± 0.13 ^c^	1.16 ± 0.24 ^ab^	0.010	0.000	0.120
*Mfn1*	1.00 ± 0.36 ^ab^	1.35 ± 0.47 ^a^	0.84 ± 0.42 ^b^	0.85 ± 0.33 ^b^	0.022	0.191	0.226
*Mfn2*	1.00 ± 0.21 ^b^	1.34 ± 0.32 ^a^	0.67 ± 0.18 ^c^	1.03 ± 0.12 ^b^	0.000	0.000	0.844

^1^ Values are expressed as mean with their SEM, *n* = 10/group. Labeled means in a row without a common letter differ, *p* < 0.05. ^2^ COX, Cytochrome c oxidase; Cyt C, Cytochrome C; Drp1, Dynamin-related protein 1; FIS1, fission 1; LBW, low birth weight; Mfn, mitofusin; NRF1, Nuclear respiratory factor 1; OPA1, Optic atrophy 1; PGC-1α, Peroxisome proliferator-activated receptor-gamma coactivator-1α; TFAM, Mitochondria transcriptional factor A. ^3^ NBW, normal birth weight. ^4^ LBW, low birth weight. ^5^ CON, a diet not supplemented with _L_-arginine. ^6^ Arg, a diet supplemented with 1.0% _L_-arginine.

## Data Availability

The datasets used to support the findings of this study are available from the corresponding author upon request.

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
