# Peer review of "Dietary Arginine Supplementation Improves Intestinal Mitochondrial Functions in Low-Birth-Weight Piglets but Not in Normal-Birth-Weight Piglets"

_antioxidants, 2021, doi:10.3390/antiox10121995_

Round 1

Reviewer 1 Report

The paper is interesting. Some minor comments are done in the attached pdf file.

General comments.

The main concern is the raising of the piglets at initial temperature of 31°C, what is lower than the critical lower temperature of piglets and could impaired LBW piglets by increasing homeothermic stress and thus oxydation. This question should be carefully adressed.

Another point is the stress induced by the separation from mother at d4. This could have different effect on LBW and NBW piglets.

Material and method.

L90-97: clarify the initial LW selection parameters. 

Results.

Table 2: provide RADG results as well.

Figure 2: some statistical data are required because a photography is not an evidence. Furthermore their quality is low.

Discussion.

L340-355: in my opinion, this part of the discussion is weak. The ratio Lys:Arg provide no further information than Arg alone.

 L382: this assertion and reference is related to yeast. So, it is out of context.

Reviewer 2 Report

The manuscript of Zhang et al. presents a continuation of a previous study of the group. The data shown give valuable information extending the knowledge on the previously studied issue. In the manuscript there are some aspects which need to be improved before the publication:

- In the data presented in the tables, the SEM (or SD) shall be given for each value. Without that it is hard to assess the extent of changes, if the scatter within particular groups is not known.

- The method for mPTP assessment shall be described in more details, as the reference only to the producer does not allow to easily find what the method really measures (the opening of mPTP or level of mPTP proteins? Which proteins?)

- The authors shall check if the results for mRNA levels of tight junction proteins are correctly presented, as the results for Occludin and Claudin look opposite to what was shown in the previous paper by the authors on the same model. If necessary, it shall be corrected or briefly commented what could be the reason of the differences.

Round 2

Reviewer 1 Report

Dear authors,

thank you for your answers.

I agree with the reply about the temperatures the piglets were exposed to.

Concerning the stress induced by early weaning, you should however be state in the paper that no sequencial samplings were performed in the course of the study (f.e. plasma cortisol,  villi height... measured at d4). So, from a theoretical point of view, it could not be excluded that some effects observed was due to higher stress in LBW piglets and not to Arg. At d4 villi height could have been higher in LBW, and the stress could have induced lower villi development in CON LBW.  

L90: in my opinion, a paradox persists: "within 1.0 ± SD" means that some NBW piglets weighed less than 1 kg (what is your definition of LBW piglets)!
